# Cancer Therapy Guided by Mutation Tests: Current Status and Perspectives

**DOI:** 10.3390/ijms222010931

**Published:** 2021-10-10

**Authors:** Svetlana N. Aleksakhina, Evgeny N. Imyanitov

**Affiliations:** 1Department of Tumor Growth Biology, N.N. Petrov Institute of Oncology, 197758 Saint-Petersburg, Russia; abyshevasv@gmail.com; 2Department of Medical Genetics, St.-Petersburg Pediatric Medical University, 194100 Saint-Petersburg, Russia

**Keywords:** mutations, genetic testing, lung cancer, melanoma, colorectal cancer, targeted therapy, TMB, HRD, MSI-H

## Abstract

The administration of many cancer drugs is tailored to genetic tests. Some genomic events, e.g., alterations of EGFR or BRAF oncogenes, result in the conformational change of the corresponding proteins and call for the use of mutation-specific compounds. Other genetic perturbations, e.g., HER2 amplifications, ALK translocations or MET exon 14 skipping mutations, cause overproduction of the entire protein or its kinase domain. There are multilocus assays that provide integrative characteristics of the tumor genome, such as the analysis of tumor mutation burden or deficiency of DNA repair. Treatment planning for non-small cell lung cancer requires testing for EGFR, ALK, ROS1, BRAF, MET, RET and KRAS gene alterations. Colorectal cancer patients need to undergo KRAS, NRAS, BRAF, HER2 and microsatellite instability analysis. The genomic examination of breast cancer includes testing for HER2 amplification and PIK3CA activation. Melanomas are currently subjected to BRAF and, in some instances, KIT genetic analysis. Predictive DNA assays have also been developed for thyroid cancers, cholangiocarcinomas and urinary bladder tumors. There is an increasing utilization of agnostic testing which involves the analysis of all potentially actionable genes across all tumor types. The invention of genomically tailored treatment has resulted in a spectacular improvement in disease outcomes for a significant portion of cancer patients.

## 1. Introduction

Laboratory genetic techniques achieved reasonable compatibility with a daily clinical practice as early as in 1980s, mainly due to the invention and rapid implementation of polymerase chain reaction (PCR) [1] and Sanger sequencing [2]. The first medical applications of DNA analysis involved the diagnosis of hereditary disorders (including familial cancer syndromes) [3], PCR-based detection of bacteria and viruses [4], genetic testing of various disease-specific rearrangements in leukemia patients [5] and HLA genotyping for organ donors and recipients [6]. Although dozens of genes mutated in cancer were known by the end of XX century, none of these alterations had clear actionability, and therefore DNA analysis of solid tumors was not a part of clinical oncology until relatively recently. The first medically relevant cancer DNA tests emerged due to chance. The development of first-generation low-weight EGFR inhibitors (EGFRi), gefitinib and erlotinib, relied on frequent overexpression of this receptor in diverse tumor types, especially in non-small cell lung cancer (NSCLC). Despite the expectations, only a few cancer patients included in the clinical trials responded to EGFRi; a posteriori analysis of responders revealed drug-sensitizing mutations in the NSCLC tissue, which were unknown at the time of the drug’s development [7]. Similarly, crizotinib was historically developed as a MET inhibitor, and its remarkable efficacy towards ALK-rearranged NSCLCs was established only after this compound entered clinical trials [8]. The first clinical investigations of anti-EGFR therapeutic antibodies were limited to colorectal carcinomas (CRCs) positive for EGFR expression by immunohistochemistry (IHC); subsequent analysis of disease outcomes revealed no relevance of EGFR status in tumor response, however, RAS mutations emerged as a strong predictor of tumor resistance to cetuximab or panitumumab [9,10].

There are several mutation tests integrated in the therapeutic decision-making process (Table 1). Some mutations, e.g., alterations in EGFR or BRAF genes, render conformational change of the corresponding protein, thus providing a rationale for the development of mutation-specific drugs. There are also genetic alterations which are accompanied by dramatic overexpression of the oncogenic enzyme, but which generally do not result in the altered interaction of its kinase domain with other molecules; the examples include ALK, ROS and RET translocations or MET exon 14 skipping mutations. In addition, some predictive DNA tests provide integrative characteristics of the cancer genome, e.g., they indicate at high tumor antigenicity due to an excessive number of mutations or druggable deficiency of DNA repair. This paper provides a brief overview on the role of genetic predictive assays in cancer treatment.

## 2. Conventional Mutation Tests in Major Cancer Types

### 2.1. Non-Small Cell Lung Cancer (NSCLC)

There are two major NSCLC histological categories, squamous cell lung cancer and non-squamous cell lung cancer. Actionable mutations are known only for non-squamous NSCLC; therefore, squamous carcinomas are currently not subjected to genetic testing [11,12].

EGFR drug-sensitizing mutations occur in 10–20% of lung adenocarcinomas in non-Asians and in approximately half of Asian NSCLCs (Figure 1). In contrast to many other actionable mutations, which are usually characteristic for more than one cancer type, EGFR alterations are highly specific for lung cancer, particularly for tumors obtained from females and/or non-smokers. There are two common EGFR mutations, ex19del and L858R, with the former being significantly more drug-sensitive than the latter [13]. In addition, the analysis of exons 18–21 may result in the detection of several rare mutations, which are sensitive to conventional EGFRi and include substitutions in codons 709, 719 and 861, as well as in-frame insertions in exon 19 [14,15,16]. There are ongoing clinical trials on novel tyrosine kinase inhibitors (TKIs), which target EGFR with insertions in exon 20 [17]. Amivantamab, a bi-specific EGFR/MET antibody, has been recently approved for the treatment of NSCLC carrying insertions in exon 20 of the EGFR gene [18,19].

ALK rearrangements are detected in approximately 5% of non-squamous NSCLCs, while the occurrence of ROS1 and RET translocations is around 2%. These gene fusions have increased prevalence in young NSCLC patients and, similarly to EGFR and ALK, are typical for female and smoking-unrelated tumors [20,21,22,23]. NSCLCs carrying druggable rearranged kinases demonstrate spectacular benefit from targeted therapies: several studies involving metastatic NSCLC cases produced median overall survival estimates well above five years, with some categories of patients approaching close to ten years survival threshold [24,25,26].

MET exon 14 skipping mutations result in increased MET protein half-life and, consequently, activated MET signaling. These mutations have a prevalence of around 2–2.5% in non-selected NSCLCs, but their frequency in elderly patients is several times higher [27]. A crizotinib study involving MET-mutated NSCLCs was clinically successful but did not result in the approval of this indication, while MET-specific inhibitor capmatinib recently received authorization for this category of NSCLCs [28,29,30].

Guidelines for NSCLC testing include the analysis of BRAF codon 600 substitutions. These mutations are detected in 1.5% of NSCLCs and are druggable by a combination of BRAF and MEK inhibitors [12].

Activating mutations in RAS genes are detected in approximately 30% of non-squamous NSCLCs. The spectrum of nucleotide substitutions is distinct between smokers and non-smokers [31]. The development of specific inhibitors of mutated RAS is highly complicated due to the small size of the protein and its high affinity to GTP. For the time being, efficient antagonists have been developed only for the KRAS G12C mutant. KRAS G12C substitution is characteristic for smoking-related NSCLCs, accounting for approximately one out of six NSCLCs in this category of patients. The first clinical trials on KRAS G12C inhibitors provided highly satisfactory results [32,33] and resulted in the approval of sotorasib [34]. The remaining RAS mutations are not druggable. However, many laboratories utilize RAS testing for NSCLC, as the presence of the RAS mutation allows reliable exclusion of other actionable mutations [12].

HER2 activating mutations occur in less than 2% of NSCLCs. These tumors can be managed by several investigational anti-HER2 drugs [35].

The analysis of all genes mentioned above is mandatory for proper NSCLC management. This procedure is technically complicated because most of the NSCLC patients are diagnosed at an advanced disease stage and can provide only tiny biopsy material for the test. The use of next generation sequencing (NGS) allows simultaneous assessment of all NSCLC-specific druggable mutations, therefore, this technique is being increasingly utilized for NSCLC analysis. There are some PCR-based pipelines which allow cost-efficient and comprehensive NSCLC testing for actionable mutations and gene fusions [12].

### 2.2. Colorectal Cancer (CRC)

CRC was the first tumor type for which molecular testing became a mandatory part of the clinical management of metastatic disease. Virtually all CRCs demonstrate activation of the RAS/RAF/MEK signaling cascade (Figure 1). This upregulation can be attributed either to the stimuli produced by membrane tyrosine kinase receptors (EGFR, or, significantly less frequently, HER2) or to the mutations in KRAS, NRAS or BRAF oncogenes [36,37]. The administration of anti-EGFR therapeutic antibodies has a potential for clinical efficacy only in the absence of other detectable mutations, which activate RAS/RAF/MEK signaling. Consequently, the use of cetuximab or panitumumab is permitted only for tumors lacking mutations in KRAS and NRAS genes. The frequency of KRAS mutations approaches 50%, while the prevalence of NRAS mutations in CRCs usually falls within 5–7%. RAS-testing in CRC is relatively complicated, because it involves the analysis of exons 2, 3 and 4 for each gene [38]. False-negative results of RAS testing, which may be attributed to the failure of tumor cell dissection or poor performance of DNA assays, possess significant risks for the patient, as EGFR inhibition in RAS-mutated tumors may facilitate the disease’s progression [39]. Comprehensive RAS analysis may require some extra time. There are clinical studies which propose to begin the treatment from a conventional therapeutic scheme and then to add anti-EGFR therapeutic antibodies starting from the second cycle, i.e., after obtaining reliable RAS-negative test results. It has been demonstrated that this delay does not worsen treatment outcomes [40]. In addition to mutations involving codons 12, 13, 59, 61 and 146, there are rare recurrent RAS substitutions occurring in other positions. Most of them, although not all, are associated with RAS activation and appear to be functionally equivalent to the hot-spot mutations listed above [41].

In contrast to NSCLC, KRAS G12C substitutions have relatively low frequency in CRC [42]. Administration of KRAS G12C inhibitors to CRC patients produced less encouraging results as compared to NSCLC, however, this option continues to be investigated in clinical trials [32]. KRAS G12C mutations are common in MUTYH-driven hereditary CRCs, which are characterized by excessive tumor mutation burden, high lymphocyte infiltration and, consequently, responsiveness to immune therapy [42]. Some studies suggest that it is feasible to perform germ-line MUTYH testing in all CRC patients, whose tumors carry the KRAS G12C substitution [43].

BRAF V600E mutations occur in 5–10% of CRCs. They are associated with a highly aggressive disease course [44]. Down-regulation of mutated BRAF kinase in CRC results in the feedback activation of the EGFR receptor [45]. Clinical trials utilizing single-agent BRAF V600E inhibitors failed, while the dual use of BRAF- and EGFR-targeted therapies provides satisfactory clinical results [46].

Approximately 2% of CRCs are driven by HER2 amplification and overexpression. These tumors are responsive to HER2-targeted therapies [47].

Some CRCs are characterized by so-called high-level microsatellite instability (MSI-H), which is caused by the deficiency in DNA mismatch repair (MMR). These tumors accumulate a significant number of alterations in mono- and dinucleotide repetitive sequences; therefore, the detection of MSI-H can be performed by the analysis of the length of appropriate microsatellite markers. This procedure requires electrophoretic separation of DNA fragments or NGS analysis, which are not always accessible in conventional morphological laboratories. IHC staining for MMR proteins is considered a reasonable substitute for MSI-H testing, as the MMR-deficient (MMR-D) phenotype is usually accompanied by the loss of MLH1/PMS2 or MSH2/MSH6 expression. MSI-H in young and/or familial CRC cases suggests the presence of Lynch syndrome, therefore, germ-line DNA testing should be offered to these patients. MSI-H in sporadic tumors is usually associated with somatic inactivation of the MLH1 gene by methylation of its promoter region; it is highly characteristic for elderly patients and is often accompanied by the BRAF V600E mutation. Metastatic MSI-H CRCs can be managed by administration of inhibitors of immune checkpoints [48,49,50].

### 2.3. Breast Cancer (BC)

All BCs require the evaluation of the HER2 status. HER2 is amplified and overexpressed in a quarter of BCs. Its testing is usually performed via IHC followed by FISH analysis of ambivalent cases, although some opinion leaders call for the upfront examination of HER2 DNA status [51]. The invention of HER2-targeted therapies resulted in a dramatic change of outcomes for HER2-driven cancers: while HER2-amplified BCs had a notoriously poor prognosis when treated in an adjuvant or metastatic setting by conventional chemotherapy, the development of a spectrum of therapeutic HER2 antagonists converted this BC subtype into a relatively well-manageable disease [52].

Approximately 15–40% of BCs carry PIK3CA activating mutations. These mutations result in the up-regulation of PI3K-driven survival pathways and render the resistance of tumor cells to various therapies. Alpelisib has recently been approved in combination with fulvestrant for the treatment of hormone receptor-positive HER2-negative PIK3CA-mutated BCs. This drug does not prolong the response to endocrine therapy in PIK3CA wild-type BCs. Alpelisib is not a mutation-specific drug in terms of its interaction with PI3K; it also down-regulates the normal version of this enzyme. Hence, the PIK3CA mutation test helps to discriminate between tumors, which are indeed supported by PI3K activation, and cancers that are characterized by the involvement of other signaling pathways. PI3K activity is essential for the cellular uptake of glucose, therefore patients receiving alpelisib have a risk of developing hyperglycemia [50].

An inherited mutation in BRCA1 or BRCA2 genes results in the occurrence of 5–8% of BCs. Tumor development involves somatic inactivation of the remaining BRCA1/2 allele, therefore BRCA1/2-driven BCs are characterized by a selective deficiency in DNA repair by homologous recombination (HRD). BRCA1/2-associated cancers demonstrate pronounced sensitivity to platinum compounds and PARP inhibitors (PARPi) [53,54].

### 2.4. Other Cancer Types 

More than 50% of skin melanomas carry activating mutations in the BRAF oncogene (Figure 1). Vemurafenib, dabrafenib and encorafenib were developed to target the most common alteration in BRAF, i.e., V600E substitution. These drugs are generally active against some other mutations affecting BRAF codon 600, particularly V600K. There are also some rare genetic alterations located in the vicinity of codon 600, which demonstrate varying sensitivity to BRAF inhibitors. Inhibition of mutated BRAF results in compensatory activation of MEK kinases, and, therefore, treatment of BRAF-mutated melanomas, always involves the combination of BRAF- and MEK-targeted drugs [55]. NRAS testing can be used for the validation of negative results of a BRAF mutation analysis, given the mutually exclusive nature of these mutations. There is a significant overlap of DNA tests utilized for NSCLC, CRC and melanoma patients (Figure 2). Approximately 15% of mucosal and acral melanomas carry activating mutations in the KIT receptor, with some of them rendering sensitivity to imatinib and nilotinib [56,57].

Sarcomas demonstrate very specific patterns of genetic aberrations: in fact, the differential diagnosis between various sarcoma subtypes is now almost entirely based on the identification of characteristic gene fusions, or some other peculiar genetic events. Unfortunately, virtually all these sarcoma-specific gene alterations are not druggable, although a few noticeable exceptions exist [58,59]. For example, the majority of gastrointestinal tumors (GISTs) contain imatinib-sensitive mutations either in exons 9, 11, 13 and 17 of the KIT oncogene (approximately 70% of cases), or, significantly less frequently, in exons 12, 14 and 18 of the PDGFRA receptor (<5%) [60,61]. In addition, there are specific imatinib-resistant mutations (PDGFRA D842V (exon 18); KIT D816V (exon 17)), which are detected in about 10% of GISTs. PDGFRA D842V substitutions are particularly common; they demonstrated sensitivity towards the recently approved drug, avapritinib [62]. Inflammatory myofibroblastic tumors often carry ALK rearrangements or, less frequently, gene fusions involving other receptor tyrosine kinases [63]. NTRK1/2/3 translocations are particularly characteristic for infantile fibrosarcomas and occur at some frequency in other sarcoma types [64]. A small subset of clear-cell sarcomas carries the BRAF V600E mutation [65].

Thyroid carcinomas may arise from follicular cells (papillary, follicular, poorly differentiated or anaplastic subtypes) or parafollicular cells C-cells (medullary subtypes) [66]. More than a half of papillary thyroid cancers are driven by BRAF mutations affecting codon 600; these tumors are sensitive to the inhibitors of mutated BRAF [67]. Up to 20% of papillary malignancies carry activating RET fusions. Medullary carcinomas account for approximately 5% of thyroid tumors. Approximately a quarter of these tumors are hereditary and develop due to germ-line mutational activation of RET kinase [68]. Two thirds of sporadic medullary thyroid cancers carry a somatic mutation in the RET oncogene. RET-driven thyroid cancers can be efficiently managed by RET inhibitors [69].

Cholangiocarcinoma is an aggressive tumor type, which is poorly manageable by conventional cytotoxic therapy [70]. Intrahepatic biliary cancers carry FGFR2 gene fusions at a rate of 10–20%. These cancers can be treated with an FGFR inhibitor, pemigatinib [71]. Another 10–20% of intrahepatic cholangiocarcinomas are characterized by mutations in IDH1 or IDH2 genes [72]. These mutations result in the accumulation of 2-hydroxyglutarate and, consequently, accumulation of epigenetic alterations [73]. An IDH1/2-targeted drug, ivosidenib, was evaluated in a randomized phase III study involving 185 patients with pretreated IDH1/2-mutated cholangiocarcinoma. Although statistically significant improvement of progression-free survival has been documented (p < 0.0001), the absolute difference between the experimental and the placebo arm was low (2.7 months versus 1.4 months) [74]. A small subset of gallbladder cancers is characterized by the presence of BRAF codon 600 substitutions. Promising clinical activity of the combination of dabrafenib and trametinib was observed in a phase II study involving 43 patients [75]. It is notable that studies on colorectal cancer demonstrated the feasibility of adding anti-EGFR therapy to BRAF inhibitors in order to prevent feedback activation of EGFR receptor [46]. Gallbladder carcinomas share gastrointestinal origin with colorectal malignancies; therefore, it is logical to anticipate that the same principle can be applied to BRAF-mutated cholangiocarcinomas. The experience of the combined inhibition of BRAF and EGFR is limited to a single case report, which demonstrated a complete response of heavily pretreated BRAF-mutated gallbladder cancer to vemurafenib, dabrafenib and irinotecan [76].

Approximately 20% of urinary tract cancers are druggable because of the presence of point activating mutations in the FGFR3 gene. FGFR inhibitor pemigatinib demonstrated good clinical activity in this category of tumors [77]. Interestingly, the efficacy of this compound was mainly limited to patients who experienced drug-induced hyperphosphatemia [78].

There are several major cancer types, e.g., esophageal, pancreatic, kidney and cervical carcinomas, among others, which rarely carry druggable genetic alterations and, therefore, are not routinely subjected to mutation testing in a clinical setting. Mutations in RAS genes (KRAS, NRAS and HRAS) are apparently the most common pan-cancer activating genetic events, as they occur at high frequency in pancreatic, colorectal, lung, skin and many other types of malignancies. With the exception of KRAS G12C substitution, there is no therapeutic compound capable of inhibiting mutated RAS genes. RAS up-regulation results in the activation of MEK kinase, however, RAS-mutated tumor cells escape MEK inhibition by autophagy. The combined use of MEK inhibitors with an autophagy antagonist, hydroxychloroquine (Plaquenil), resulted in the shrinkage of RAS-mutated pancreatic and colorectal tumors in several case reports [79,80,81]. These observations require validation in properly designed clinical trials and may eventually result in major changes in the landscape of cancer diagnostics and treatment.

## 3. Integrative Tests

The tumor mutation burden (TMB) reflects the total number of mutations per genome. It is especially high in cancers induced by excessive carcinogenic exposure, e.g., lung cancer or melanoma, and in tumors deficient for particular DNA repair pathways. High TMB results in increased tumor antigenicity and, consequently, responsiveness to immune therapy. Although there are some technical issues related to the methodology of TMB measurement and definition of the threshold, there is a good consistency across the studies estimating the predictive value of TMB in different tumor types. The correlation between high TMB and increased benefit from therapy has been observed for immune checkpoint inhibitors (ICIs) [82,83]. The FDA approved the use of pembrolizumab for the treatment of tumors with ≥ 10 mutations per megabase [84,85]. The evaluation of TMB absolutely requires the use of NGS, either for the analysis of the entire exome or for sequencing of selected representative fragments of tumor DNA. ESMO guidelines provide the list of tumor types for which TMB evaluation is recommended as a routine practice [11]. The history of heavy smoking may be considered as a surrogate for TMB in NSCLC patients [86].

MSI-H is in fact a subcategory of the high-TMB phenotype. Microsatellite-unstable tumors have a characteristic pattern of mutations, which are caused by the deficiency of the MMR module for DNA repair. In contrast to the determination of TMB in microsatellite-stable tumors, the analysis of MSI-H does not necessarily require NGS. MSI-H status can be determined by molecular analysis using a panel of standard microsatellite markers, or by IHC [49]. Some tumors carry the hypermutator phenotype due to a deficiency of other components of maintenance of genomic stability. MUTYH-associated colorectal carcinomas have an excessive amount of G:C > T:A substitutions due to a deficiency in base excision repair [42]. The hypermutator phenotype is also characteristic for tumors with inactivation of DNA polymerase epsilon (POLE) [87].

Some malignancies, particularly ovarian and triple-negative breast carcinomas, are characterized by the deficiency of DNA repair by homologous recombination. The HRD (BRCAness) phenotype can be caused by inactivation of BRCA1/2 genes or some other genes belonging to the same pathway, and is associated with tumor responsiveness to platinum compounds, mitomycin C, PARP inhibitors (PARPi), etc. HRD-deficient cancers cannot efficiently repair DNA double-strand breaks and, therefore, accumulate multiple copy number abnormalities across the genome. The most frequent cause of this chromosomal instability is a biallelic inactivation of BRCA1/2 genes in hereditary cancers [88,89,90]. BRCA1 and BRCA2 germ-line mutations predispose to breast, ovarian and possibly stomach malignancies [88,90,91]. In addition, BRCA2 pathogenic alleles are associated with increased risk of prostate and pancreatic carcinomas [92,93]. Interestingly, not all cancers arising in BRCA1/2 mutation carriers are characterized by somatic inactivation of the remaining BRCA1/2 allele; therefore, it is advisable to supplement BRCA1/2 germ-line testing with the BRCA1/2 loss-of-heterozygosity (LOH) analysis of the tumor tissue [94,95]. There are NGS-based assays aimed at evaluating the genome-wide status of chromosomal instability and identifying a subset of tumors sensitive to platinum-based or PARPi therapy. Sporadic cancers with the BRCAness phenotype generally demonstrate lower sensitivity to BRCA1/2-specific drugs, as compared to hereditary BRCA1/2-driven malignancies. There are continuing attempts to simplify HRD testing and to adapt it to the daily clinical use [89,96].

## 4. Agnostic versus Tissue-Specific Targets

Targeted therapy is based on the assumption that the mere presence of actionable vulnerability in cancer cell is associated with the responsiveness to a properly selected drug. Although being generally true, this statement is an oversimplification and may need some adjustment for a tissue-specific tumor context (Figure 3). For example, BRAF-mutated melanomas, which do not express significant amounts of the EGFR receptor, can be managed by a combination of BRAF and MEK inhibitors. In contrast to melanomas, BRAF-driven colorectal cancers require EGFR inhibition to prevent an activation of the compensatory signaling cascade, and are usually treated by combined administration of anti-EGFR antibodies and antagonists of mutated BRAF [97]. IDH1/2 inhibitors are active in IDH1/2-mutated acute myeloid leukemia, but demonstrate limited efficacy in glioblastomas carrying IDH1/2 genetic alterations [98,99,100]. The clinical activity of inhibitors of KRAS G12C is evidently higher in lung versus colorectal cancers [32]. PIK3CA inhibitors demonstrated satisfactory activity in breast carcinomas, but showed low efficacy in non-breast cancer clinical trials; it is essential to acknowledge that the BC trials involved the combination of PIK3CA with endocrine therapy, while other studies utilized single-agent PIK3CA inhibition [101,102,103].

There are several markers which received agnostic status by FDA approval [85]. High-level microsatellite instability, which occurs mainly in colorectal, endometrial and gastric cancers, but is rare in other tumor types, is an indication for the use of immune therapy [84,104]. Similarly, a high tumor mutation burden can be utilized as an agnostic marker for the administration of immune checkpoint inhibitors [84]. NTRK1/2/3 gene rearrangements, which are found at a reasonable frequency in pediatric tumors but are exceptionally rare in adults, are associated with the tumor’s responsiveness to entrectinib or larotrectinib [105,106,107,108]. ALK, ROS1 and RET rearrangements, although not formally classified as agnostic markers, occur in multiple tumor types and render sensitivity to appropriate inhibitors [63,109,110,111,112]. HER2 amplification accompanied by gene overexpression may also be considered as an example of a more or less agnostic druggable event [113,114,115].

## 5. Multigene Testing for the Choice of Therapy

The concept of agnostic markers led to the development of multigene panels, which cover the entire spectrum of all known drug targets and aim to support the choice of an effective therapy [85]. While it is probably beyond the debate that the curated knowledge on the status of all coding regions of the genome and its integrative characteristics (TMB score, HRD status) is generally desirable for every tumor case, the widespread use of NGS is currently limited by high costs and, to a lesser extent, significant turn-around time of the analysis. Besides the economical consideration, there are intrinsic conceptual difficulties related to the utilization of multigene testing.

There are only a few known targets (EGFR, BRAF, MET, HER2, ALK, ROS1, RET, NTRK, BRCA1/2, MSI-H, high TMB, HRD) which indeed can be used for therapy guidance in a relatively straightforward way [85]. Most of the currently utilized NGS panels include an excessive number of genes with, at best, uncertain predictive value [116,117,118]. For example, the I-PREDICT trial interpreted TP53 mutations as a reason for the administration of antiangiogenic therapy, despite the scarcity of supporting evidences for this decision [116]. There are several clinical investigations which included patients with exhausted standard treatment options and involved the administration of drugs based on the results of agnostic NGS tests. All these trials observed clinical responses in a subset of patients, with many instances of tumor shrinkage attributed to the above-listed well-known actionable alterations. Overall, it is difficult to make a quantitative assessment of the results of these trials, as they involved diverse cancer types and considered both targets with a very high probability of tumor response and last-hope treatment options with clearly insufficient levels of evidence [116,117,118].

There are ESMO recommendations which inform clinical specialists on the utilization of NGS in clinical oncology. These recommendations suggest the routine use of NGS only for selected tumor types (NSCLC, ovarian cancer, prostate cancer, cholangiocarcinoma) and only for a relatively small spectrum of genes which have sufficiently proven predictive value. It is stated that the use of extended gene panels or enlargement of the spectrum of analyzed tumor types increases the probability of finding a druggable target only by a small extent, and should be utilized either for investigational purposes or with explicit knowledge on the expected cost–benefit ratio [11,119].

## 6. Conclusions and Perspectives

The development of DNA-based tests and corresponding mutation-specific drugs led to dramatic improvement of outcomes in a subset of oncological patients. However, the overall number of tumors which have clearly actionable genetic alterations is relatively low, therefore, the majority of subjects with cancer do not receive gene-tailored treatment. Virtually all major cancer types have already been subjected to systematic exome sequencing studies, but this effort revealed only a modest number of novel potentially druggable mutated genes [120,121,122]. Apparently, a single-target drug selection is gradually reaching its plateau, and many opinion leaders call for the development of integrative therapeutic approaches. There are attempts to identify universal vulnerabilities of transformed cells, e.g., activation of particular signaling pathways or some general metabolic dependencies [123,124]. Some studies suggest consideration of the entire spectrum of molecular alterations in each given tumor and the individual customization of the combination of targeted drugs for every cancer patient [85,116]. Mutation profiling covers only a part of tumor molecular characteristics, hence, there are efforts to integrate genomic, transcriptomic, proteomic and metabolomic profiling into a single decision-making process [117,125]. Although molecular medicine is definitely guiding the development of novel cancer treatments, it may provide a breakthrough only when coupled with innovations in the clinical management of oncological patients. For example, there is sound evidence suggesting that not only surgery but even systemic therapy can be efficient only in patients with a relatively small tumor burden [126]. Consequently, early diagnosis and screening of malignant diseases will continue to play an utmost role in the fight against cancer [127]. The invention of immune checkpoint inhibitors boosted clinical research on neoadjuvant therapy, therefore, an increasing number of potentially operable patients experience systemic intervention before the surgery [128,129]. Monitoring of the tumor’s molecular evolution during the treatment course is becoming a part of the clinical routine [130]. Interaction between surgeons, medical oncologists, radiologists, pathologists and molecular biologists is a critical factor for success in cancer care.

## Figures and Tables

**Figure 1 ijms-22-10931-f001:**
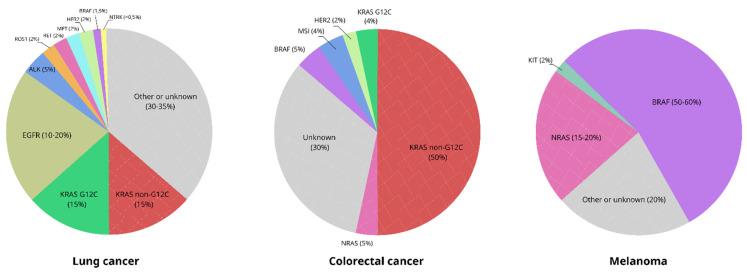
Spectrum of actionable mutations in lung carcinomas, colorectal cancers and skin melanomas.

**Figure 2 ijms-22-10931-f002:**
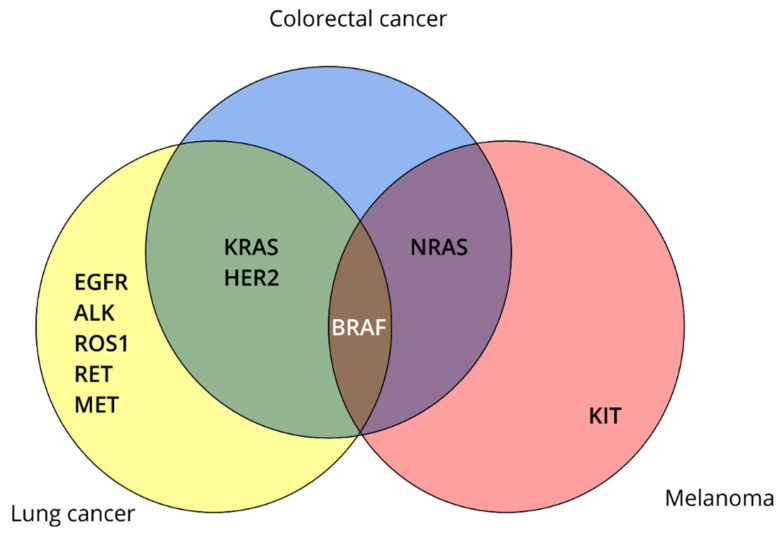
Common predictive mutations in major cancer types. Some mutation tests are more or less organ-specific, while other DNA assays are helpful in several categories of cancer patients.

**Figure 3 ijms-22-10931-f003:**
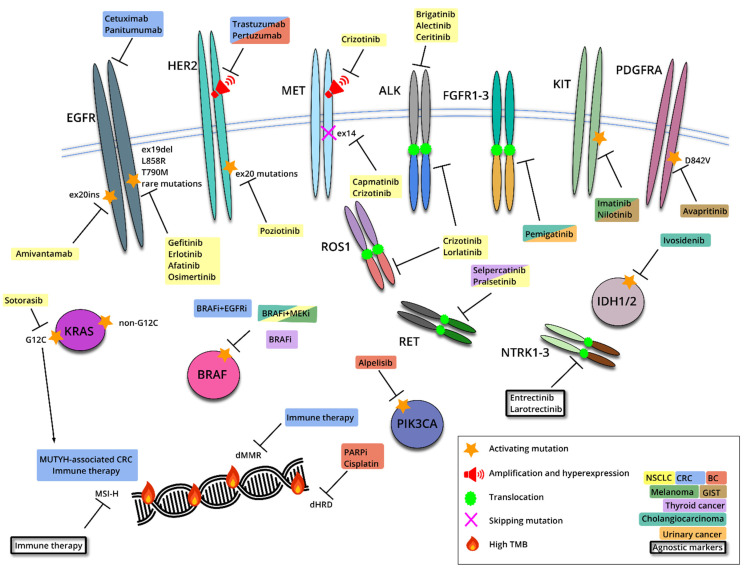
Examples of tissue-specific and agnostic drug targets. Most known druggable mutations occur in tyrosine kinase membrane receptors, however, some of these molecules can relocate to other cellular compartments due to mutational events. Some targets and drugs are generally relevant only to a single tumor category (e.g., EGFR mutations in NSCLC), other indications include several cancer types (e.g., use of BRAF inhibitors for BRAF V600E mutated tumors), and a few targeted therapies are considered agnostic. The T-shaped lines illustrate the blockade of particular targets by various drugs.

**Table 1 ijms-22-10931-t001:** Examples of predictive mutation tests utilized in clinical oncology.

Cancer Type	Genetic Lesions
Lung cancer	EGFR, BRAF, MET, HER2, KRAS G12C mutations
	ALK, ROS1, RET rearrangements
Colorectal cancer	KRAS/NRAS mutations (exclusion of patients from anti-EGFR therapy)
	BRAF mutations
	HER2 amplifications
	Microsatellite instability
Breast cancer	HER2 amplifications
	PIK3CA mutations
	BRCA1/2 germ-line pathogenic variants
Melanomas	BRAF, KIT mutations
Sarcomas	GIST: KIT, PDGFRA (GIST)
	Inflammatory myofibroblastic tumors: ALK and other gene rearrangements
	Infantile fibrosarcomas and other sarcomas: NTRK1/2/3 rearrangements
	Clear-cell sarcomas: BRAF mutations
Ovarian cancer	BRCA1/2 germ-line pathogenic variants
	HRD
Stomach cancer	HER2 amplifications
Glioblastomas	IDH1/2 mutations
Cholangiocarcinomas	IDH1/2, BRAF mutations
Endometrial cancer	Microsatellite instability
Prostate cancer	BRCA1/2 germ-line pathogenic variants
Pancreatic cancer	BRCA1/2 germ-line pathogenic variants
Thyroid cancer	RET mutations and rearrangements, BRAF mutations
Urinary tract carcinoma	FGFR3 mutations
Agnostic markers	Microsatellite instability, NTRK1/2/3 rearrangements, TMB

See comments in the text.

## Data Availability

Not applicable.

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
