# Peer review of "Cancer Therapy Guided by Mutation Tests: Current Status and Perspectives"

_ijms, 2021, doi:10.3390/ijms222010931_

Round 1

Reviewer 1 Report

I have read the article carefully, the text seems clear and exhaustive to me. The issues were addressed well and well described.

My suggestions are: - better specify the genetic risk in subjects carrying the Brca mutation - write a small paragraph on the mutated genes involved in colorectal cancer

Author Response

Question: better specify the genetic risk in subjects carrying the Brca mutation -

Answer: We have inserted this information in the paragraph 3 of the chapter 3.

BRCA1 and BRCA2 germ-line mutations predispose to breast, ovarian and possibly stomach malignancies [88,90,91]. In addition, BRCA2 pathogenic alleles are associated with increased risk of prostate and pancreatic carcinomas [92,93].

Question: write a small paragraph on the mutated genes involved in colorectal cancer

Answer: Please notice that the chapter 2.2 is devoted to colorectal cancer. To make it more visible, we incorporated “Colorectal cancer” in the heading (in addition to abbreviations “CRC”).

Reviewer 2 Report

This is a informative review:

Figure is fine. Maybe one Table (not busy one) would be helpful too.

 1. There are missing area on sarcoma (soft tissue sarcoma has a lot of diagnostic mutations and translocations), brain tumor(IDHs and MGMT stories), and blood neoplasms (most advanced area probably).

2. Some modification on title OR include those in the review.

3. Area of uncertainty would be helpful for readers.

4. Unmet area such as pancreas cancer could be addressed, too.

Author Response

Question: Maybe one Table (not busy one) would be helpful too.

Answer: We have incorporated Table 1.

Question: There are missing area on sarcoma (soft tissue sarcoma has a lot of diagnostic mutations and translocations), brain tumor(IDHs and MGMT stories), and blood neoplasms (most advanced area probably).

Answer: We have added a paragraph describing sarcomas. Gastrointestinal stromal tumors are now incorporated in this paragraph.

Sarcomas demonstrate very specific patterns of genetic aberrations: in fact, the differential diagnosis between various sarcoma subtypes is now almost entirely based on the identification of characteristic gene fusions or some other peculiar genetic events. Unfortunately, virtually all these sarcoma-specific gene alterations are not druggable, although a few noticeable exceptions exist [58, 59]. For example, the majority of gastrointestinal tumors (GISTs) contain imatinib-sensitive mutations either in exons 9, 11, 13 and 17 of KIT oncogene (approximately 70% of cases), or, significantly less frequently, in exons 12, 14 and 18 of PDGFRA receptor (<5%) [60, 61]. In addition, there are specific imatinib-resistant mutations (PDGFRA D842V (exon 18); KIT D816V (exon 17)), which are detected in about 10% of GISTs. PDGFRA D842V substitutions are particularly common; they demonstrated sensitivity towards recently approved drug avapritinib [62]. Inflammatory myofibroblastic tumors often carry ALK rearrangements or, less frequently, gene fusions involving other receptor tyrosine kinases [63]. NTRK1/2/3 translocations are particularly characteristic for infantile fibrosarcomas and occur at some frequency in other sarcoma types [64]. A small subset of clear-cell sarcomas carry BRAF V600E mutation [65]. 

IDH1/2 mutations in glioblastomas are mentioned in the chapter 4 and in the Table 1. MGMT gene is not affected by mutations, so, to our opinion, its description  is outside the scope of this review. We intentionally did mention hematological malignancies, as it is an entirely separate field; our review is devoted to cancer, while leukemia etc. is similar but still especial category of oncological diseases.  

Question: Some modification on title OR include those in the review.

Answer: Please see the comments above.

Questions: Area of uncertainty would be helpful for readers; Unmet area such as pancreas cancer could be addressed, too.

Answer: We have added an additional paragraph to the chapter 2.4 .

There are several major cancer types, e.g., esophageal, pancreatic, kidney, cervical, etc. carcinomas, which rarely carry druggable genetic alterations and, therefore, are not routinely subjected to mutation testing in clinical setting. Mutations in RAS genes (KRAS, NRAS and HRAS) are apparently the most common “pan-cancer” activating genetic events, as they occur at high frequency in pancreatic, colorectal, lung, skin and many other types of malignancies. With exception of KRAS G12C substitution, there is no therapeutic compounds capable of inhibiting mutated RAS genes. RAS up-regulation results in activation of MEK kinase, however RAS-mutated tumor cells escape MEK inhibition by autophagy. Combined use of MEK inhibitors with an autophagy antagonist hydroxychloroquine (Plaquenil) resulted in shrinkage of RAS-mutated pancreatic and colorectal tumors in several case reports [79-81]. These observations require validation in properly designed clinical trials and may eventually result in major changes in the landscape of cancer diagnostics and treatment.